# $\mathbb{Z}_2 \times \mathbb{Z}_2$ Equivariant Quantum Neural Networks: Benchmarking against Classical Neural Networks

Zhongtian Dong [1,*,†], Marçal Comajoan Cara [2,†], Gopal Ramesh Dahale [3,†], Roy T. Forestano [4,†], Sergei Gleyzer [5,†], Daniel Justice [6,†], Kyoungchul Kong [1,†], Tom Magorsch [7,†], Konstantin T. Matchev [4,†], Katia Matcheva [4,†] and Eyup B. Unlu [4,†]

1 Department of Physics & Astronomy, University of Kansas, Lawrence, KS 66045, USA; kckong@ku.edu
2 Department of Signal Theory and Communications, Polytechnic University of Catalonia, 08034 Barcelona, Spain; marcal.comajoan@estudiantat.upc.edu
3 Indian Institute of Technology Bhilai, Kutelabhata, Khapri, District-Durg, Chhattisgarh 491001, India; gopald@iitbhilai.ac.in
4 Institute for Fundamental Theory, Physics Department, University of Florida, Gainesville, FL 32611, USA; roy.forestano@ufl.edu (R.T.F.); matchev@ufl.edu (K.T.M.); matcheva@ufl.edu (K.M.); eyup.unlu@ufl.edu (E.B.U.)
5 Department of Physics & Astronomy, University of Alabama, Tuscaloosa, AL 35487, USA; sgleyzer@ua.edu
6 Software Engineering Institute, Carnegie Mellon University, 4500 Fifth Avenue, Pittsburgh, PA 15213, USA; dljustice@sei.cmu.edu
7 Physik-Department, Technische Universität München, James-Franck-Str. 1, 85748 Garching, Germany; tom.magorsch@tum.de
* Correspondence: cdong@ku.edu
† These authors contributed equally to this work.

**Abstract:** This paper presents a comparative analysis of the performance of Equivariant Quantum Neural Networks (EQNNs) and Quantum Neural Networks (QNNs), juxtaposed against their classical counterparts: Equivariant Neural Networks (ENNs) and Deep Neural Networks (DNNs). We evaluate the performance of each network with three two-dimensional toy examples for a binary classification task, focusing on model complexity (measured by the number of parameters) and the size of the training dataset. Our results show that the $\mathbb{Z}_2 \times \mathbb{Z}_2$ EQNN and the QNN provide superior performance for smaller parameter sets and modest training data samples.

**Keywords:** quantum computing; deep learning; quantum machine learning; equivariance; invariance; supervised learning; classification; particle physics; Large Hadron Collider

**MSC:** 81P68; 68Q12



## 1. Introduction

The rapidly evolving convergence of machine learning (ML) and high-energy physics (HEP) offers a range of opportunities and challenges for the HEP community. Beyond simply applying traditional ML methods to HEP issues, a fresh cohort of experts skilled in both areas is pioneering innovative and potentially groundbreaking approaches. ML methods based on symmetries play a crucial role in improving data analysis as well as expediting the discovery of new physics [1,2]. In particular, classical Equivariant Neural Networks (ENNs) exploit the underlying symmetry structure of the data, ensuring that the input and output transform consistently under the symmetry [3]. ENNs have been widely used in various applications including deep convolutional neural networks for computer vision [4], AlphaFold for protein structure prediction [5], Lorentz equivariant neural networks for particle physics [6], and many other HEP applications [7–11].

Meanwhile, the rise of readily available noisy intermediate-scale quantum computers [12] has sparked considerable interest in using quantum algorithms to tackle high-energy physics problems. Modern quantum computers boast impressive quantum volume

and are capable of executing highly complex computations, driving a collaborative effort within the community [13,14] to explore their applications in quantum physics, particularly in addressing theoretical challenges in particle physics. Recent research on quantum algorithms for particle physics at the Large Hadron Collider (LHC) covers a range of tasks, including the evaluation of Feynman loop integrals [15], simulation of parton showers [16] and structure [17], development of quantum algorithms for helicity amplitude assessments [18], and simulation of quantum field theories [19–24].

An intriguing prospect in this realm is the emerging field of quantum machine learning (QML), which harnesses the computational capabilities of quantum devices for machine learning tasks. With classical machine learning algorithms already proving effective for various applications at the LHC, it is very natural to explore whether QML can enhance these classical approaches [25–34]. In recent years, significant development has been made in their quantum counterparts, Equivariant Quantum Neural Networks (EQNNs) [35–39].

In this paper we benchmark the performance of EQNNs against various classical and/or non-equivariant alternatives for three two-dimensional toy datasets, which exhibit a $\mathbb{Z}_2 \times \mathbb{Z}_2$ symmetry structure. Such patterns often appear in high-energy physics data, e.g., as kinematic boundaries in the high-dimensional phase space describing the final state [40,41]. By a clever choice of the kinematic variables for the analysis, these boundaries can be preserved in projections onto a lower-dimensional feature space [42–46]. For example, one can form various combinations of possible invariant mass for the generic decay chain considered in Ref. [44], $D \rightarrow jC \rightarrow j\ell_n^{\pm}B \rightarrow j\ell_n^{\pm}\ell_f^{\mp}A$, where Particles $A$, $B$, $C$, and $D$ are hypothetical particles in new physics beyond the standard model of masses $\{m_A, m_B, m_C, m_D\}$, while the corresponding standard model decay products consist of a jet $j$, a "near" lepton $\ell_n^{\pm}$, and a "far" lepton $\ell_f^{\pm}$. The two-dimensional (bivariate) distribution $\frac{d^2\Gamma}{dR_{ij}dR_{kl}}$ shows distributions similar to those in Figures 1–3, where $R_{ij} = \frac{m_i^2}{m_j^2}$ is the mass square ratio. Symmetric, anti-symmetric, or non-symmetric structures provide information of particle masses involved in the cascade decays.

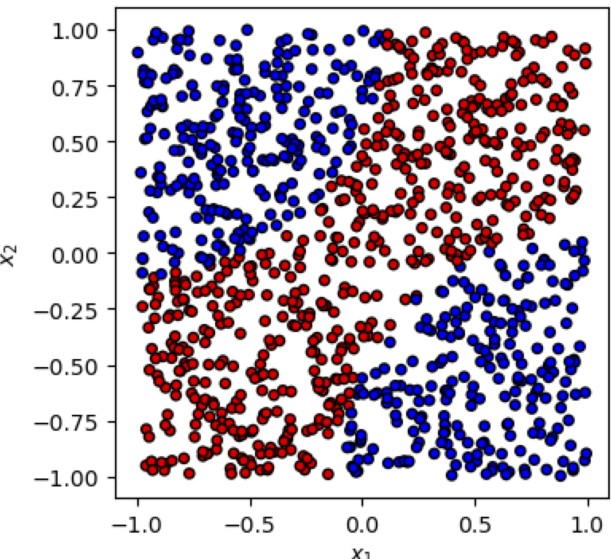

**Figure 1.** Pictorial illustration of the first dataset used in this study—the symmetric case (1).

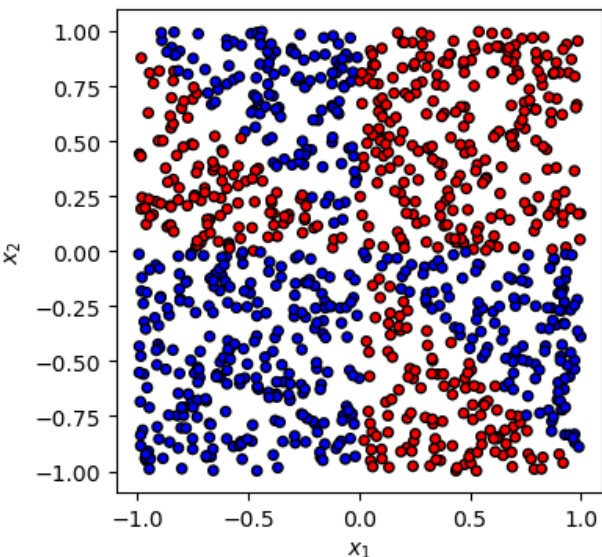

**Figure 2.** Pictorial illustration of the second dataset used in this study—the anti-symmetric case (4).

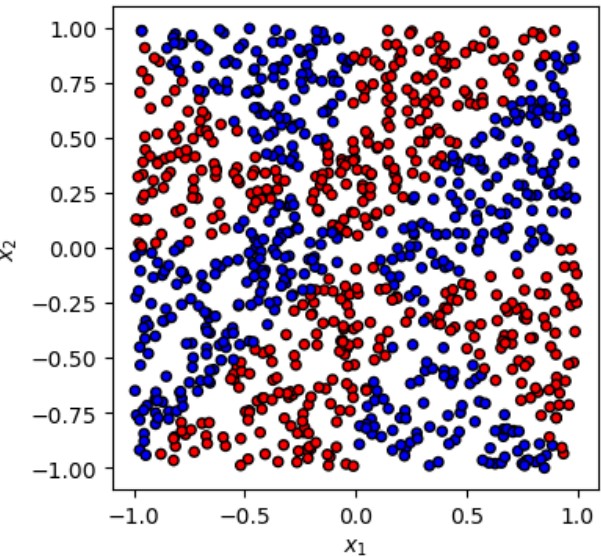

**Figure 3.** Pictorial illustration of the third dataset used in this study—the fully anti-symmetric case (6).

In this study, we consider simplified two-dimensional datasets that mimic the data arising in such projections. This setup allows us to focus on the comparison between different methods, avoiding unnecessary issues that may arise when dealing with actual particle physics simulation data such as sampling statistics, parton distribution functions, unknown particle mass spectrum, unknown width, detector effects, etc. We explore EQNNs and benchmark them against classical neural network models. We find that the variational quantum circuits learn the data better with the smaller number of parameters and the smaller training dataset compared to their classical counterparts.

## 2. Dataset Description

In all three examples, we consider two-dimensional data $(x_1, x_2)$ on the unit square $(-1 \leq x_i \leq 1)$. The data points belong to two classes: $y = +1$ (blue points) and $y = -1$ (red points).

(i) Symmetric case:

In the first example (Figure 1), the labels are generated by the function

$$
\begin{aligned}
y(x_1, x_2) \;=\; & 2H\!\left(R - \sqrt{(x_1 + 1)^2 + (x_2 - 1)^2}\right) \\
& +\; 2H\!\left(R - \sqrt{(x_1 - 1)^2 + (x_2 + 1)^2}\right) - 1,
\end{aligned}
\tag{1}
$$

where $H(x)$ is the Heaviside step function and for definiteness we choose $R = 1.1$. The function (1) respects a $\mathbb{Z}_2 \times \mathbb{Z}_2$ symmetry, where the first $\mathbb{Z}_2$ is given by a reflection about the $x_1 = x_2$ diagonal

$$
x_1 \to x_2, \qquad x_2 \to x_1, \qquad y \to y,
\tag{2}
$$

while the second $\mathbb{Z}_2$ corresponds to a reflection about the $x_1 = -x_2$ diagonal

$$
x_1 \to -x_2, \qquad x_2 \to -x_1, \qquad y \to y.
\tag{3}
$$

This $\mathbb{Z}_2 \times \mathbb{Z}_2$ example was studied in Ref. [37] and we shall refer to it as the symmetric case since the $y$ label is invariant.

(ii) Anti-symmetric case:

The second example is illustrated in Figure 2. The labels are generated by the function

$$
\begin{aligned}
y(x_1, x_2) = & \; H(-x_1)H(-x_2) + H(-x_1)H(x_2)H(x_1 + x_2) \\
& - H(x_1)H(x_2) + H(x_1)H(-x_2)H(x_1 + x_2).
\end{aligned}
\tag{4}
$$

The first $\mathbb{Z}_2$ is still realized as in (2). However, this time, the labels are flipped under a reflection along the $x_1 = -x_2$ diagonal:

$$
x_1 \to -x_2, \qquad x_2 \to -x_1, \qquad y \to -y,
\tag{5}
$$

which is why we shall refer to this case as anti-symmetric.

(iii) Fully anti-symmetric case:

The last example is depicted in Figure 3. The labels are generated by the function

$$
\begin{aligned}
y(x_1, x_2) = & \; H(x_1)H(x_2)(2H(x_1 - x_2) - 1) + H(-x_1)H(-x_2)(2H(x_2 - x_1) - 1) \\
& + H(-x_1)H(x_2)H(x_1 + x_2)\left(2H\!\left(R - \sqrt{(x_1 + 1)^2 + (x_2 - 1)^2}\right) - 1\right) \\
& + H(-x_1)H(x_2)H(-x_1 - x_2)\left(1 - 2H\!\left(R - \sqrt{(x_1 + 1)^2 + (x_2 - 1)^2}\right)\right) \\
& + H(x_1)H(-x_2)H(x_1 + x_2)\left(1 - 2H\!\left(R - \sqrt{(x_1 - 1)^2 + (x_2 + 1)^2}\right)\right) \\
& + H(x_1)H(-x_2)H(-x_1 - x_2)\left(2H\!\left(R - \sqrt{(x_1 - 1)^2 + (x_2 + 1)^2}\right) - 1\right),
\end{aligned}
\tag{6}
$$

where $H(x)$ is the Heaviside step function and for definiteness we choose $R = 1$. In this case, the labels are flipped under both reflections along the $x_1 = -x_2$ diagonal as well as the $x_1 = x_2$ diagonal, which is why we shall refer to this case as fully anti-symmetric. As we will see later, it is straightforward to incorporate both symmetric and anti-symmetric properties in variational quantum circuits, while it is not obvious how to consider the anti-symmetric case in the classical neural networks.

## 3. Network Architectures

To assess the importance of embedding the symmetry in the network, and to compare the classical and quantum versions of the networks, we study the performance of the

following four different architectures: (i) Deep Neural Network (DNN), (ii) Equivariant Neural Network (ENN), (iii) Quantum Neural Network (QNN), and (iv) Equivariant Quantum Neural Network (EQNN). In each case, we adjust the hyperparameters to ensure that the number of network parameters is roughly the same.

(i)  Deep Neural Networks:
     In our DNN, for the symmetric (anti-symmetric) case, we use one (two) hidden layer(s) with four neurons. For both types of classical networks, we use the softmax activation function, Adam optimizer, and a learning rate of 0.1. We use the binary cross-entropy for both the DNN and ENN.

(ii)  Equivariant Neural Networks:
     A given map $f : x \in X \rightarrow f(x) \in Y$ between an input space $X$ and an output space $Y$ is said to be equivariant under a group $G$ if it satisfies the following relation:

$$f(g_{\text{in}}(x)) = g_{\text{out}}(f(x)), \tag{7}$$

where $g_{\text{in}}$ ($g_{\text{out}}$) is a representation of a group element $g \in G$ acting on the input (output) space. In the special case when $g_{\text{out}}$ is the trivial representation, the map is called invariant under the group $G$, i.e., a symmetry transformation acting on the input data $x$ does not change the output of the map. The goal of ENNs, or equivariant learning models in general, is to design a trainable map $f$ which would always satisfy Equation (7). In tasks where the symmetry is known, such equivariant models are believed to have an advantage in terms of the number of parameters and training complexity. Several studies in high-energy physics have attempted to use classical equivariant neural networks [6,47–50]. Our ENN model utilizes four $\mathbb{Z}_2 \times \mathbb{Z}_2$ symmetric copies for each data point, which are fed into the input layer, followed by one equivariant layer with three (two) neurons and one dense layer with four (four) neurons in the symmetric (anti-symmetric) case.

(iii)  Quantum Neural Networks:
     For the QNN, we utilize the one-qubit data-reuploading model [51], as shown in Figure 4, with depth four (eight) for the symmetric (anti-symmetric and fully anti-symmetric) case, using the angle embedding and three parameters at each depth. This choice leads to a similar number of parameters as in the classical networks. We use the Adam optimizer and the loss

$$L_{QNN} = y(1 - |\langle \psi | O_1 | \psi \rangle|)^2 + (1 - y)(1 - |\langle \psi | O_2 | \psi \rangle|)^2 \tag{8}$$

for any choice of two orthogonal operators $O_1$ and $O_2$ (see Ref. [52] for more details.). In this paper, we use

$$O_1 = \begin{pmatrix} 1 & 0 \\ 0 & 0 \end{pmatrix}, \qquad O_2 = \begin{pmatrix} 0 & 0 \\ 0 & 1 \end{pmatrix}. \tag{9}$$

for all three datasets considered in this paper.

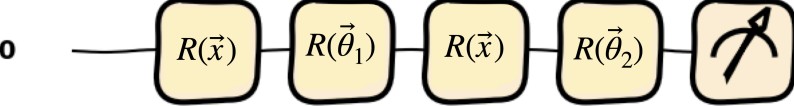

**Figure 4.** Illustration of the quantum circuit used for QNN at depth 2. This circuit is repeated up to depth four (five) times with different parameters for the symmetric (anti-symmetric and fully anti-symmetric) case. The data points $\vec{x} = (x_1, x_2, 0)$ are loaded via angle embedding with rotation gates, followed by another rotation $R(\vec{x}) = R_Z(0)R_Y(x_2)R_Z(x_1)$ with arbitrary angle parameters.

(iv) Equivariant Quantum Neural Networks.

In EQNN models, symmetry transformations acting on the embedding space of input features are realized as finite-dimensional unitary transformations $U_g$, $g \in G$. Consider the simplest case where one trainable operator $U(\theta, x)$ acts on a state $|\psi\rangle$: $U(\theta, x) |\psi\rangle$. If for a symmetry transformation $U_g$, the condition

$$U(\theta, x) U_g |\psi\rangle = U_g U(\theta, x) |\psi\rangle, \tag{10}$$

is satisfied, then the operator $U$ is equivariant, i.e., the equivariant gate should commute with the symmetry. In general, the $U_g$ operators on the two sides of Equation (10) do not necessarily have to be in the same representation but are often assumed so for simplicity. The output of a QNN is the measurement of the expectation value of the state with respect to some observable $O$. If the gates are equivariant and we apply some symmetry transformation $U_g$, then this is equivalent to measuring the observable $U_g^\dagger O U_g$. Hence, if $O$ commutes with the symmetry $U_g$, the model as a whole would be invariant under $U_g$, which is the case in our symmetric example. Otherwise the model is equivariant, as in our anti-symmetric example.

Our EQNN uses the two-qubit quantum circuit depicted in Figure 5 for depth 1. This circuit is repeated five (ten) times with different parameters for the symmetric (anti-symmetric and fully anti-symmetric). The two $R_Z$ gates embed $x_1$ and $x_2$, respectively. The $R_X$ gates share the same parameter ($\theta_1$) and the $R_{ZZ}$ gate uses another parameter ($\theta_2$). The invariant model (for the symmetric case) uses the same observable $O$ for both classes in the data. In the anti-symmetric case, we use two different observables $O_1$ and $O_2$ that correspond to each label. They transform into one another under reflection $g_r$, i.e., $U_{g_r}^\dagger O_1 U_{g_r} = O_2$.

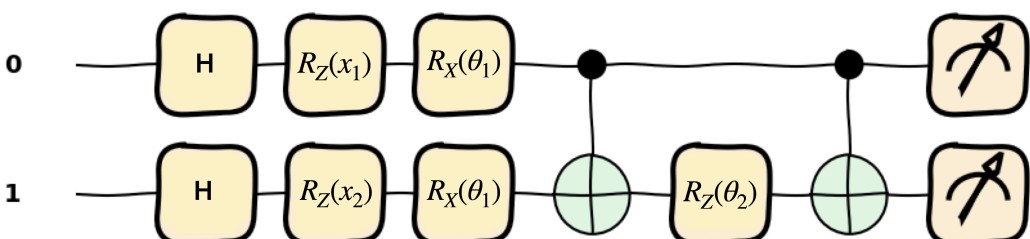

**Figure 5.** Illustration of the quantum circuit used for EQNN at depth 1. This circuit is repeated five (ten) times with different parameters for the symmetric (anti-symmetric and fully anti-symmetric) case. The data points $(x_1, x_2)$ are loaded via angle embedding with two $R_Z$ gates, $R_Z(x_1)$ and $R_Z(x_2)$. The remaining circuits are parameterized by $R_X(\theta_1)$ and $R_Z(\theta_2)$.

In the symmetric case, we use binary cross-entropy loss, assuming the true label $y$ is either 0 or 1,

$$L_{EQNN}^{symm} = y \log \left( |\langle \psi | O | \psi \rangle | \right) + (1 - y) \log \left( 1 - |\langle \psi | O | \psi \rangle | \right). \tag{11}$$

The observables $O$ and the reflection $U_{g_r}$ along $x_1 = -x_2$ are defined as follows:

$$O = \frac{1}{4} \begin{pmatrix} 1 & 1 & 1 & 1 \\ 1 & 1 & 1 & 1 \\ 1 & 1 & 1 & 1 \\ 1 & 1 & 1 & 1 \end{pmatrix}, \qquad U_{g_r} = \begin{pmatrix} 0 & 0 & 0 & 1 \\ 0 & 0 & 1 & 0 \\ 0 & 1 & 0 & 0 \\ 1 & 0 & 0 & 0 \end{pmatrix}. \tag{12}$$

In the anti-symmetric and fully anti-symmetric cases, we used the same loss as in QNN

$$L_{EQNN}^{anti-symm} = y(1 - |\langle \psi | O_1 | \psi \rangle |)^2 + (1 - y)(1 - |\langle \psi | O_2 | \psi \rangle |)^2. \tag{13}$$

For the anti-symmetric case $O_1$ ($O_2$) is the observable corresponding to $y = 1$ ($y = 0$)

$$O_1 = \frac{1}{4} \begin{pmatrix} 1 & 1 & 1 & -1 \\ 1 & 1 & 1 & -1 \\ 1 & 1 & 1 & -1 \\ -1 & -1 & -1 & 1 \end{pmatrix}, \qquad O_2 = \frac{1}{4} \begin{pmatrix} 1 & -1 & -1 & -1 \\ -1 & 1 & 1 & 1 \\ -1 & 1 & 1 & 1 \\ -1 & 1 & 1 & 1 \end{pmatrix}. \tag{14}$$

For the fully anti-symmetric case, we use another set of observables, so one will transform into the other with reflection along any of the two diagonals. They are given as follows:

$$O_1 = \frac{1}{4} \begin{pmatrix} 1 & -1 & 1 & 1 \\ -1 & 1 & -1 & 1 \\ 1 & -1 & 1 & -1 \\ 1 & 1 & -1 & 1 \end{pmatrix}, \qquad O_2 = \frac{1}{4} \begin{pmatrix} 1 & 1 & -1 & 1 \\ 1 & 1 & -1 & -1 \\ -1 & -1 & 1 & 1 \\ 1 & -1 & 1 & 1 \end{pmatrix}. \tag{15}$$

Since it is anti-symmetric with respect to each of the diagonals, the result is invariant if both reflections are applied. It is difficult to build a classical equivariant neural network using these anti-symmetries since classical equivariant models are built based on the assumption that the target is invariant under certain transformations. When discussing the theory of classical equivariant machine learning models, the models that transform non-trivially under the symmetry group are often discussed mathematically but rarely implemented in code. For our classical model on partially anti-symmetric data, we only implemented the invariant part of the symmetry ($\mathbb{Z}_2$) and ignored the anti-symmetric portion of the data. While it may not be impossible to consider such asymmetric cases in classical neural networks, implementation can be quite involved.

On the other hand, it is straightforward to build quantum equivariant models. For this purpose, we would only need to exploit the transformation properties of the observables. If one observable transforms to the other under the transformation of interest (reflection along the diagonal in this case), then measurement made on one observable is equivalent to the measurement of the other observable given the transformed input.

We can consider equivariant quantum models with anti-symmetric transformation from the point of view of representation theory. The fully invariant (symmetric) case can be considered as the model transform under the trivial representation of the group, where all the transformations defined by the group do not change the output of the model. The asymmetric (either anti-symmetric or fully anti-symmetric) cases that we considered here can be interpreted as transforms under some other (one-dimensional) representation of the group, where some transformations change the output of the model to its opposite value, while other transformations do not change the output.

## 4. Results

The left panels in Figure 6 show the receiver operating characteristic (ROC) curves for each network with $N_{\text{train}} = 200$ and $N_{\text{test}} = 2000$ samples for the symmetric (top), anti-symmetric (middle), and fully anti-symmetric (bottom) dataset. The results for the DNN, ENN, QNN, and EQNN are shown in (green, dotted), (yellow, dotdashed), (red, dashed), and (blue, solid), respectively. As expected, networks with an equivariance structure (EQNN and ENN) improve the performance of the corresponding networks (QNN and DNN) without the symmetry. We also observe that quantum networks perform better than the classical analogs. In the legends, numerical values followed by network acronyms represent the number of parameters used for each network. For the symmetric example, the EQNN uses only 10 parameters; thus, for fair comparison, we constructed the other networks with $\mathcal{O}(10)$ parameters as well. For the anti-symmetric example, we use 20 parameters for the EQNN.

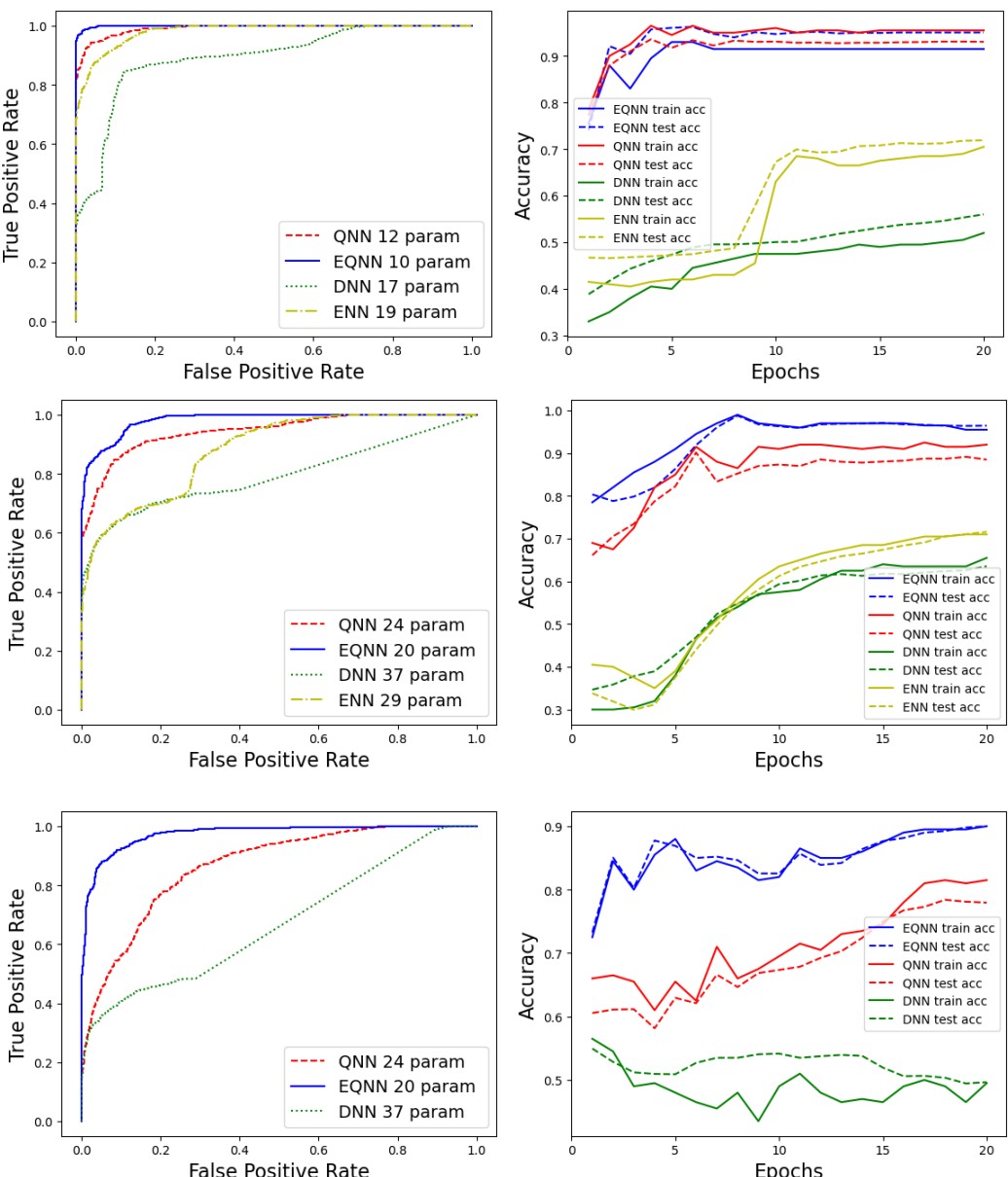

**Figure 6.** ROC (**left**) and accuracy (**right**) curves for the symmetric (**top**), anti-symmetric (**middle**), and fully anti-symmetric (**bottom**) example.

The evolution of the accuracy during training and testing is shown in the right panels of Figure 6. The accuracy converges faster (after only 5 epochs) for the QNN and EQNN in comparison to their classical counterparts (10–20 epochs). The same color-scheme is used, but this time, solid curves represent training accuracy, while dashed curves show test accuracy.

To further quantify the performance of our quantum networks, in Figure 7, we show the AUC (Area under the ROC Curve) as a function of the number of parameters (left panels) with a fixed size of the training data ($N_{\text{train}} = 200$), and as a function of the number of training samples (right panels) with a fixed number of parameters ($N_{\text{params}} = 20$). The top, middle, and bottom panels show results for the symmetric, anti-symmetric, and fully anti-symmetric dataset. As the number of parameters increases, the performance of all networks improves. All AUC values become similar when $N_{\text{params}} \approx 20$ ($N_{\text{params}} \approx 40$) for the symmetric (anti-symmetric) case. As shown in the bottom panels, the performances of all networks become comparable to each other for both examples once the size of the training data reaches $\sim$400,

except for the fully anti-symmetric case. We observe that from the top panel to the bottom panel, the relative improvement from the QNN to the EQNN grows, indicating the importance of symmetry implementation on the network. Similar relative improvement exists from the DNN to the QNN, emphasizing the importance of quantum algorithms. Note that the ENN curves are missing in the bottom panel of both Figures 6 and 7. This is due to the non-trivial implementation of the anti-symmetric property in classical ENNs.

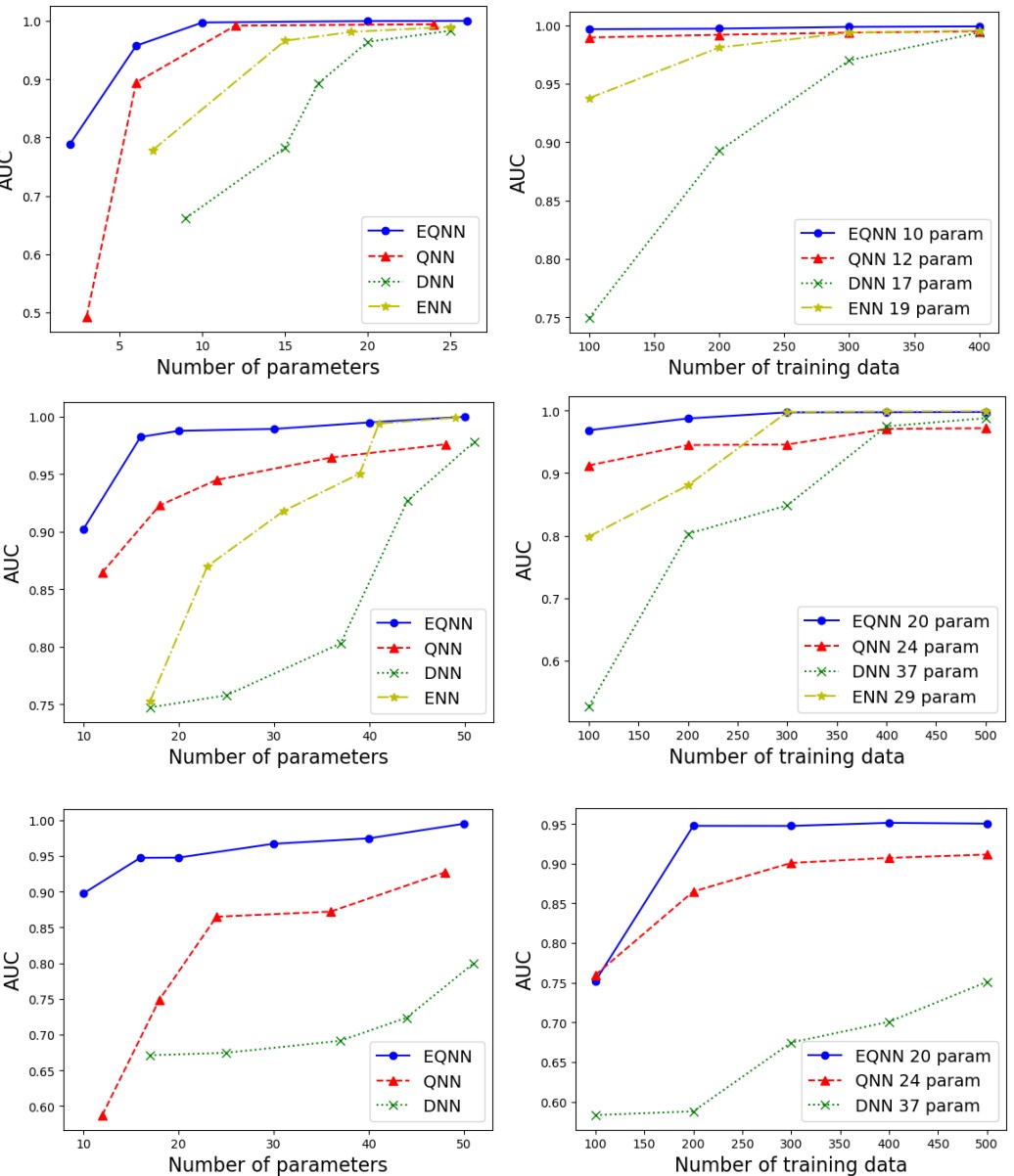

**Figure 7.** AUC as a function of the number of parameters (**left**) for fixed $N_{\text{train}} = 2000$ and $N_{\text{test}} = 200$, and as a function of $N_{\text{train}}$ (**right**) with a fixed number of parameters as shown in the legend, for the symmetric (**top**), anti-symmetric (**middle**), and fully anti-symmetric example (**bottom**).

Finally Table 1 shows the accuracy of the DNN for the fully anti-symmetric dataset. The different rows and columns represent different choices of the number of parameters and the number of training samples, respectively. These numbers are compared against those in right-bottom panel of Figure 7. The EQNN achieves 0.95 accuracy with 20 parameters and 200 training samples, while the DNN requires more parameters and/or more training samples.

**Table 1.** Accuracy of DNN for the fully anti-symmetric dataset. The different rows (columns) represent different choices of the number of parameters (the number of training samples).

| $N_{\text{params}} \backslash N_{\text{train}}$ | 100 | 200 | 300 | 400 | 500 | 600 | 700 | 800 | 900 |
|---|---|---|---|---|---|---|---|---|---|
| 105 | 0.764 | 0.855 | 0.879 | 0.963 | 0.973 | 0.981 | 0.981 | 0.982 | 0.988 |
| 85 | 0.669 | 0.743 | 0.804 | 0.953 | 0.951 | 0.978 | 0.986 | 0.946 | 0.981 |
| 67 | 0.587 | 0.722 | 0.695 | 0.946 | 0.886 | 0.9632 | 0.975 | 0.944 | 0.980 |
| 51 | 0.624 | 0.655 | 0.856 | 0.926 | 0.908 | 0.876 | 0.846 | 0.974 | 0.986 |
| 37 | 0.596 | 0.696 | 0.639 | 0.782 | 0.747 | 0.816 | 0.849 | 0.922 | 0.952 |

## 5. Conclusions

In this paper, we examined the performance of Equivariant Quantum Neural Networks and Quantum Neural Networks, compared against their classical counterparts, Equivariant Neural Networks and Deep Neural Networks, considering two toy examples for a binary classification task. Our study demonstrates that EQNNs and QNNs outperform their classical counterparts, particularly in scenarios with fewer parameters and smaller training datasets. This highlights the potential of quantum-inspired architectures in resource-constrained settings. This point has been emphasized in a similar study recently in Ref. [35], which showed that an EQNN outperforms the non-equivariant one in terms of generalization power, especially with a small training set size. We note a more significant enhancement in the performance of an EQNN and QNN compared to an ENN and DNN, particularly evident in the anti-symmetric example rather than the symmetric one. This underscores the robustness of quantum algorithms. The code used for this study is publicly available at https://github.com/ZhongtianD/EQNN/tree/main (accessed on 7 March 2024).

While our current study has primarily focused on an EQNN with discrete symmetries, it is crucial to acknowledge the significant role that continuous symmetries, such as Lorentz symmetry or gauge symmetries, play in particle physics. In our future research, we aim to compare an EQNN with continuous symmetries against classical neural networks. Exploring more complex datasets with high-dimensional features is another direction we plan to pursue. However, handling such examples would necessitate an increase in the number of network parameters, prompting an investigation into related issues like overparameterization, barren plateaus, and others.

**Author Contributions:** Conceptualization, Z.D.; methodology, M.C.C., G.R.D., Z.D., R.T.F., S.G., D.J., K.K., T.M., K.T.M., K.M. and E.B.U.; software, Z.D.; validation, M.C.C., G.R.D., Z.D., R.T.F., T.M. and E.B.U.; formal analysis, Z.D.; investigation, M.C.C., G.R.D., Z.D., R.T.F., T.M. and E.B.U.; resources, Z.D., K.T.M. and K.M.; data curation, G.R.D., S.G. and T.M.; writing—original draft preparation, Z.D.; writing—review and editing, S.G., D.J., K.K., K.T.M. and K.M.; visualization, Z.D.; supervision, S.G., D.J., K.K., K.T.M. and K.M.; project administration, S.G., D.J., K.K., K.T.M. and K.M.; funding acquisition, S.G. All authors have read and agreed to the published version of the manuscript.

**Funding:** This research used resources of the National Energy Research Scientific Computing Center, a DOE Office of Science User Facility supported by the Office of Science of the U.S. Department of Energy under Contract No. DE-AC02-05CH11231 using NERSC award NERSC DDR-ERCAP0025759. SG is supported in part by the U.S. Department of Energy (DOE) under Award No. DE-SC0012447. KM is supported in part by the U.S. Department of Energy award number DE-SC0022148. KK is supported in part by the US DOE DE-SC0024407. CD is supported in part by the College of Liberal Arts and Sciences Research Fund at the University of Kansas. CD, RF, EU, MCC, and TM were participants in the 2023 Google Summer of Code.

**Institutional Review Board Statement:** Not applicable.

**Data Availability Statement:** The dataset used in this analysis was sampled from Equations (1), (4) and (6).

**Conflicts of Interest:** The authors declare no conflicts of interest. The funders had no role in the design of the study; in the collection, analyses, or interpretation of data; in the writing of the manuscript; or in the decision to publish the results.

## Abbreviations

The following abbreviations are used in this manuscript:

| | |
|---|---|
| API | Application Processing Interface |
| AUC | Area Under the Curve |
| DNN | Deep Neural Network |
| ENN | Equivariant Neural Network |
| EQNN | Equivariant Quantum Neural Network |
| HEP | High-Energy Physics |
| LHC | Large Hadron Collider |
| MDPI | Multidisciplinary Digital Publishing Institute |
| ML | Machine Learning |
| NN | Neural Network |
| QML | Quantum Machine Learning |
| QNN | Quantum Neural Network |
| ROC | Receiver Operating Characteristic |

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
