# Peer review of "2 × ℤ2 Equivariant Quantum Neural Networks: Benchmarking against Classical Neural Networks"

_axioms, doi:10.3390/axioms13030188_

Round 1

Reviewer 1 Report

Comments and Suggestions for Authors

The manuscript contributes significantly to quantum computing by exploring EQNNs and benchmarking them against classical neural network models. To further underscore the importance and novelty of this research, an expanded discussion on the specific challenges addressed by EQNNs, alongside their advantages over classical models, would be beneficial. Highlighting these aspects early could help emphasize the study's relevance and innovative approach.

In terms of methodology, the experimental design is well-articulated. Yet, the manuscript would gain depth by providing more details on the selection criteria for the examples and the rationale behind the chosen model parameters (10.1109/QCE57702.2023.00033, 10.26044/ecr2023/C-16014). Such elaboration would not only aid in understanding the relevance of the experimental setup but also ensure the reproducibility of the results. Additionally, while the results support the conclusions drawn, incorporating further statistical analyses or visual representations could enhance the reader's comprehension of the performance differences between EQNNs and classical models. This approach would make the findings more accessible and underscore their significance.

The manuscript would also benefit from a broader discussion of its limitations and future research directions (10.1038/s41534-023-00710-y, 10.1109/ISBI53787.2023.10230448). Outlining potential applications of EQNNs in other domains or suggesting methods to overcome current constraints could provide valuable insights into the broader implications of the work. Moreover, ensuring that all references are present and directly relevant, including recent studies that support or challenge the findings, would enrich the manuscript's context and demonstrate its contribution to the ongoing discourse in quantum computing.

Finally, thorough proofreading to correct minor grammatical and typographical errors is recommended to enhance the manuscript's clarity and professionalism. Additionally, supplementary materials such as additional datasets, code, or detailed methodological appendices would significantly benefit readers and researchers interested in further understanding or extending the research, thereby amplifying its impact and utility in the field.

Comments on the Quality of English Language

While the manuscript is generally well-written, minor grammatical corrections and clarity enhancements could benefit a few sections. A thorough proofreading by a native English speaker is recommended to polish the manuscript.

Author Response

We thank the referee for the valuable comments and suggestions.

Please see the attached PDF, which contains our response. 

Reviewer 2 Report

Comments and Suggestions for Authors

The paper \lq $\mathbb{Z}_2 \times \mathbb{Z}_2$ equivariant quantum neural netwoorks: benchmarking against classical neural networks' by Zhongtian Dong et al is a short contribution (7 pages) aiming to compare the performance of three types of neural networks (EQQN: equivariant quantum neural networks, QNN) and their classical counterparts. This comparison is nowadays such a standard subject that the authors seem to borrow their entire abstract and introduction to Ref. [1] (see the similarity report).
The good point is that there exists a series of jupyter notebooks written by the first author available on github (with a link given in the conclusion).

The presentation is clear and the paper is worthwhile. However it is not clear what is new compared to previous work. The state of the art is not presented. The paper looks like a short note for proceedings not like a full length paper. I cannot recommend the publication in the present form.

Author Response

We thank the referee for all the comments and suggestions. 

Please see the attached PDF which contains our response. 

Reviewer 3 Report

Comments and Suggestions for Authors

In this manuscript, authors perform a comparison of performance of EQNN with the classical counterparts. The findings are interesting. Although the manuscript is not prepared well, I think it can be improved. Because it is very timely, I would recommend its publication in Axioms, after a major revision addressing the following concerns.

1- Presentation of the related works in the literature is missing. I guess this not the first time that a quantum NN is compared to classical NN?

2- What about EQNN, and Z_2 x Z_2? A comprehensive explanation of these are required.

3- It looks like it is a kind of a part of, or somehow related to a particle/high energy physics research, which makes it even more interesting. However, in its current form, the connections are missing. For example, there are two keywords “particle physics; Large Hadron Collider” and several references, which at first glance look to have nothing to do with the present manuscript. The connection / motivation should be made clear. In its current form, the manuscript looks like a lab report prepared quickly upon obtaining some results. However, it could be prepared much better.

4- To me, the major concern is the following. It is understandable that with increasing data size the performance of all the networks improves. However, how come the performance of classical networks catch up the quantum networks? There is only a small sentence “[Line 112] As the number of parameters increases, the performance of all networks improves.” The question is how and why? There is no physical explanation. Normally, one would expect the gap between the performance of quantum and classical NNs even increases. Is the overall quantum model or the quantum circuit in Fig.2 not good enough?

5- Lastly in this round, what about the angle of Z and X rotations in the quantum circuit? How to determine the angles? Are they the same every time, or should the optimal values be determined for each data set?

Author Response

(The authors gave the same response as above.)

Round 2

Reviewer 2 Report

Comments and Suggestions for Authors

The paper was much improved according to the recommendations. It can be accepted.

Reviewer 3 Report

Comments and Suggestions for Authors

The authors have improved their manuscript significantly. It can now be recommended for publication.